# Thermodynamic and Kinetic Aspects of Gold Adsorption in Micrometric Activated Carbon and the Impact of Their Loss in Adsorption, Desorption, and Reactivation Plants

**DOI:** 10.3390/ma16144961

**Published:** 2023-07-12

**Authors:** Rodrigo Martínez-Peñuñuri, Jose R. Parga-Torres, Jesus L. Valenzuela-García, Héctor J. Díaz-Galaviz, Gregorio González-Zamarripa, Alejandro M. García-Alegría

**Affiliations:** 1Department of Materials and Metallurgy, Instituto Tecnológico de Saltillo, Tecnológico Nacional de México, Saltillo 25280, Mexico; 2Department of Chemical Engineering and Metallurgy, Universidad de Sonora, Hermosillo 83260, Mexico; 3Department of Materials, Instituto Tecnológico de Monclova, Tecnológico Nacional de México, Monclova 95245, Mexico; 4Department of Chemical Biological Sciences, Universidad de Sonora, Hermosillo 83260, Mexico

**Keywords:** activated carbon, adsorption, gold, heap leaching, adsorption, desorption, and reactivation plants

## Abstract

The production and loss of fine particles of activated carbon (AC) loaded with gold in the adsorption processes is a worldwide problem, mainly due to the behavior of increasing its adsorption capacity with respect to the decrease in particle size, which becomes relevant to determine the thermodynamic and kinetic activity of the gold adsorption and the economic impact of its loss, with the escape towards the later stages of the system of adsorption, desorption, and reactivation (ADR) plants of AC. Through the adsorption of gold in a synthetic medium with sodium cyanide concentration, using different particle sizes, AC weights, and adsorption times, data were generated for analysis by three different isotherm models, resulting in a better tendency for the Freundlich isotherm, from which thermodynamic parameters of Δ*G* equal to −2.022 kcal/mol, Δ*H* equal to −16.710 kcal/mol, and Δ*S* equal to −0.049 kcal/molK were obtained, which shows that it is a spontaneous, exothermic process with a lower degree of disorder. The kinetic analysis was performed with two different models, from which the pseudo-second-order model was used due to a better tendency and displayed a behavior that leaves open the interpretation of the increase in adsorption with respect to the decrease in the AC particle size but demonstrated the importance of recovering these particles in relation to their gold concentration and the economic impact from their poor recovery, which, for this case study, amounted to ~0.3 million dollars per year.

## 1. Introduction

The typical gold mining process has multiple stages, including ore crushing (crushing and grinding), leaching, activated carbon (AC) adsorption, AC desorption, electrodeposition or precipitation with zinc dust, and gold refining [1,2]. Gold leaching is typically accomplished using a sodium cyanide (NaCN) solution; the CN ion forms an aurocyanide complex, which remains in a dissolved state within the solution [3,4]. NaCN is one of the key factors in gold dissolution; however, it does not exist in isolation from the cyanide-to-oxygen ratio derived from the Elsner equation [5,6], as follows:4Au + 8NaCN + O_2_ + 2H_2_O → 4 Na[Au(CN)_2_] + 4NaOH(1)

Metal recovery systems with AC have been widely used because they do not require treatment of the solution with a concentration of precious metals produced by leaching before recovery and because of their great versatility [7,8].

The adsorption mechanism of precious metal carbons in cyanide solutions has the following principal characteristics:Extraction is enhanced in the presence of electrolytes such as CaCl_2_ and KCl;Adsorption kinetics and equilibrium loading increase as the pH decreases;Gold adsorption increases the pH of the solution;Neutral cyanide complexes, such as Hg(CN)_2_, are strongly adsorbed regardless of the ionic strength of the solution;Adsorption is a reversible process, with a higher stripping rate for slightly different conditions;There is evidence that adsorption is dependent on the reduction potential of the system;Under most conditions, the molar ratio of loaded gold to nitrogen is 0.5:1.0, which is consistent with the presence of Au(CN)_2_;Adsorption decreases with increasing temperatures;The adsorption mechanism can be represented by the following equation:(2)$${M^{n+}+nAu(CN)}_{2}^{-}=M^{n+}{\left[{Au\left(CN\right)}_{2}^{-}\right]}_{n(ads)}$$

Kinetic analysis further indicates that the initial adsorption rate of the aurocyanide complex is faster, which decreases as it approaches equilibrium. The adsorption rate is controlled by the mass transfer of the complex to the surface of the AC. Once pseudo-equilibrium is reached, the diffusion of the complex through the micropores of carbon decreases more than through the boundary layer because of the length and complexity of the pores. The activation energy for gold adsorption is approximately 11 kJ/mol, indicating mass transport control [9,10].

The gold adsorption rate in AC can be described by a first-order velocity equation [11]:(3)$$\log C_{t}=\log C_{0}+kt$$
where *C_t_* is the gold concentration at any time *t*, *C*_0_ is the initial gold concentration, and *k* is a rate constant.

The mean consumption of AC in adsorption, desorption, and reactivation (ADR) plants is 40 g of AC per ton of ore processed. This consumption is attributed to various factors during the production of fine AC particles, resulting from the mixture of adsorption circuits (40%), carbon transfer between stages (including transfer to elution: 6%), the elution process (including regeneration transfer: 7%), and regeneration (including cooling and final sizing: 47%) [12].

It is estimated that 41% of the AC loss within the adsorption circuit (adsorption plus AC transfer between stages equals 46%) is lost in the final tailings, with only 5% recovered in the tailing safety screen [12]. The AC lost in the tailings contains gold adsorbed during the adsorption process. As this AC moves through the adsorption circuit, it remains unclear whether it releases the contained gold or transfers it to particles with lower gold charges. This is because the generated fine AC passes rapidly through a circuit with the suspension, and there is unlikely to be equilibrium or a charge of gold.

The loss of AC particles containing adsorbed gold concentrations in ADR plants has implications beyond just economic consequences. It also has a potential effect on gold extraction from heap leaching processes. In particular, the transfer of AC particles from the reconditioning piles of cyanide solutions to the heaping leaching stage could mimic “preg-robbing”. In preg-robbing, gold cyanide complexes are removed from the solution by an adsorbent compound in the ore [13,14].

The ADR industrial plants in the region employ AC with a particle size of 3.36–1.7 mm (equivalent to 6–12 meshes) in their operations. These plants typically utilize a particle capture mesh of 1.68–0.4 mm. Thus, particles with a size less than 0.4 mm inevitably escape [12].

This study demonstrates and justifies the importance of preventing the loss of AC particles. In the laboratory setting, we conducted simulations to investigate the adsorption rate and the load capacity of AC particles with a size less than 0.4 mm. Subsequently, the kinetic and thermodynamic parameters of gold adsorption were determined at the industrial level by monitoring the amount of AC (in mass) and the gold concentrations within such particles for 24 h [2].

This study highlights the importance of the efficiency of industrial metallurgical processes in gold recovery, demonstrates the impact of AC fine particle losses, and justifies the development of alternative techniques to recover AC particles. All in the context of the tendency for low concentrations of gold in new mineral deposits and the increasing costs associated with gold extraction difficulties due to its mineralogy [15,16,17,18].

## 2. Materials and Methods

### 2.1. Reagents and Materials

The adsorption procedures of batch-type AC were investigated through laboratory-scale experimentation in an aqueous medium containing NaCN with a concentration of 1000 mg/L (Meyer, Tláhuac, Mexico, Lot J1121698, Catalog 2350). The AC (Calgon Carbon DG-11 6 × 12 Lot HB184AN-4, Pittsburgh, Pennsylvani) from coconut shell-based granular activated carbons and specifically designed for gold recovery operations are made with a manufacturing process through direct activation. They are heated (approximately 500 °C in the presence of dehydrating agents) and removed or remain as tar-like residues giving to the AC a crystallographic formations, resulting in a specific area between 10 and 500 m^2^/g (and sometimes as high as 1000 m^2^/g), and exposing the carbonized material to an oxidizing atmosphere of steam, carbon dioxide and/or oxygen (air) at temperatures of 700 to 1000 °C to burn off the tar-like residues and to develop the internal pore structure, to macropores (x > 100 to 200 nm), mesopores (1.6 < x < 100 to 200 nm), and micropores (x < 1.6 nm), where x is the characteristic size [12]. The AC was prepared in fine particles, a mortar was used to achieve a reduction in size, while meshes (Wstyler Test Sieve, Mentor, OH, USA, ASTM E11 Standard) were used to screen AC and generate particle sizes of 150 μm (mesh 100, serial number 184318681), 106 μm (mesh 140, serial number 195124716), 75 μm (mesh 200, serial number 170719543), 53 μm (mesh 270, serial number 201813053), 45 μm (mesh 325, serial number 202517329), and 38 μm (mesh 400, serial number 203021146) for gold adsorption. Within 24 h, the reaction was conducted in a 600 mL beaker (Pyrex, Stoke-on-Trent, UK) and stirred using a magnetic stirrer at 700 rpm (Fisher brand model Isotemp). We used a 1000 mg/L synthetic matrix solution (HPS lot number 2132210-250) alkalized with 2.5 g of sodium hydroxide (J.T. Baker lot number Y26C83, Bohus, Sweden), reaching a pH of 10.50–11.00 (VWR pH Meter Model BP30Cl, Shanghai, China). Buffer solutions (J.T. Baker) of pH 4, 7, and 10 were used to calibrate the pH meter. The test medium was adjusted to 0.5 L of synthetic solution containing Au at 10 mg/L in contact with 0.25, 0.125, 0.05, and 0.01 g of AC in different particle sizes.

### 2.2. Laboratory Equipment

The test set and the experimental arrangement for the adsorption process were established using the following equipment, materials, and reagents: RoTap (WS Tyler RX-29 model, serial number 28390), pH meter (VWR, model B10P), pH meter (VWR, model BP30Cl), magnetic stirring grate and magnetic stirring bar (3 cm; Fisherbrand, Shanghai, China, model Isotemp), analytical balance (Ohaus, model Adventurer Pro AC64C, Pine Brook, NJ, USA), mercury thermometers scale of −10 °C–110 °C (Chemillé en Anjou, France), 1000 mL beakers (Pyrex), 10 cm diameter funnels of separation (Pyrex), and a 10 cm diameter Whatman No. 40 filter paper.

The characterization procedures involved the use of the following equipment: a Cress Melting Furnace (model C1632, serial number 910), an analytical microbalance (Mettler Toledo, model XP-6, serial number B518889533, Greifensee, Switzerland), a Becomar (Gas Extraction Hood, model CP-180, Toluca de Lerdo, Mexico), a heating plate (LabTech, model EG20B, serial number 211202G13746, Hopkinton, MA, USA), an atomic absorption spectrophotometer (Agilent Technologies, model 55AA, serial number MY21040001, Petaling Jaya, Malaysia), and scanning optical microscopes (model JSM-6610LV from JEOL, Akishima, Japan and model Pro X from Phenom, Eindhoven, The Netherlands).

### 2.3. Analytical Procedures

Various techniques were used to supplement and validate their results to determine the concentration of the element of interest in solutions as well as its concentration and characteristics in AC. The methods used were flame atomic absorption spectroscopy, fire assay, gravimetric determination, microbalance, scanning electron microscopy (SEM), and spectroscopy of dispersed energy.

#### 2.3.1. Atomic Absorption Spectroscopy

Model AA 240 FS from Agilent Technologies was used for the determination of gold in atomic adsorption monitoring solutions in AC. In addition, this instrument was employed for the determination of gold by fire assay when the detection limit (<10 mg/L of gold) was the best option for the procedure. The tests involved conducting three replicates per aliquot with a sampling time of 3 s and a reading time of 5 s per replicate in a calibration curve of 4 points [19]. The 4-point calibration curve was prepared using a gold matrix solution with traceability of a concentration of 1000 mg/L (HPS lot number 2132210-250) and was prepared using automatic micropipettes and flasks for adjustments to 0.5–1.0–1.5–3.0 ppm of gold, equaling the matrix to 1000 ppm of NaCN with a pH of 10.5–11.0.

#### 2.3.2. Fire Assay

A fire assay was used for the quantitative characterization of gold adsorbed in AC after adsorption procedures in cyanide aqueous media. The fire assay process was specifically developed for the fusion of AC particles [20] using a Cress Melting Furnace Model C1632, acid attack of the bead (the gold-silver product from the fire assay process) using nitric acid (Merck brand lot 1.00456.2510, Darmstadt, Germany) to remove silver, and gold determination using the analytical microbalance.

#### 2.3.3. SEM

The microstructure of AC was determined using SEM. Surface morphological analysis was performed through secondary electron imaging and composition analysis using backscattered electron imaging [21].

### 2.4. Adsorption Test for Thermodynamic and Kinetic Analyses

The study involved conducting experiments using AC DG-11, wherein various particle sizes were generated through mechanical rupture using a mortar as follows: 106 μm (140 mesh), 75 μm (200 mesh), 53 μm (270 mesh), 45 μm (325 mesh), and 38 μm (400 mesh), these sizes of AC represent particles that scape from the screening (0.7 to 0.8 mm) step after the reactivation process and also from the safety screen (0.4 mm) that is located at the end of the train of columns on ADR plants [12]. The adsorption process was conducted for 120 min in a 600 mL beaker at room temperature (25 °C), atmospheric pressure, and solution conductivity of 640 μS/cm, stirred using a magnetic stirrer at 750 rpm, with monitoring intervals at 1, 3, 5, 10, 15, 20, 30, 40, 50, 60, 90, and 120 min after the start of the test, taking 5 mL of solution at every interval [22,23,24,25].

The gold synthetic matrix used in the test was prepared by combining doubly distilled water with a gold matrix solution of 1000 mg/L, alkalized with 2.5 g of sodium hydroxide in beads, resulting in a pH of 10.50–11.00.

The operation conditions for each of the tests were as follows: 0.5 L of matrix solution and 10 mg/L of gold in contact with 0.010, 0.050, 0.125, and 0.250 g of AC DG-11 in different particle sizes.

Table 1 shows the operating conditions of the tests for the determination of the load capacity and adsorption rate of AC.

### 2.5. Industrial Monitoring and Sampling of Escaping Fine AC Particles

A 24 h monitoring and sampling program was conducted on Molimentales del Noroeste SA de CV from 4 November 2022 to 5 November 2022, in order to quantify, from an economic perspective, the value of gold loss on ADR systems. This included time recording and the flow of NaCN solution at the beginning and end of monitoring.

The sampling was intended to collect approximately 1 L per hour for a period of 24 h, specifically in the exit of the cyanide solution after the adsorption process in the columns and in the mesh, which aimed to retain fine AC (Calgon Carbon DG-11 6 × 12) particles. Further, 50 mL of the solution was sampled at the start of the adsorption process every hour for a period of 24 h to determine the initial gold concentration in the cyanide solution.

Duplicate samples of 1 L of the cyanide solution were collected at the exit of the process at the monitoring intervals of 2, 3, 4, 6, 8, 9, 12, 18, and 21 h. The samples collected for the 24 h monitoring interval were taken in triplicate. The intention of doubling and tripling was to ensure a sufficient amount of fine AC particles for the analysis of fire assay particles. The solution intake commenced by performing a triple rinse using the same solution of interest, eliminating any potential sources of cross-contamination.

The sampling was intended to collect approximately 1 L of solution every hour for a period of 24 h. This was specifically conducted at the exit of the cyanide solution after undergoing the adsorption process in the columns and the mesh, which is designed to capture fine AC particles. In addition, 50 mL of the solution was sampled at the start of the adsorption process every hour for 24 h to determine the initial gold concentration in the cyanide solution.

The analysis of each cyanide solution sample began with the weight determination and recording of each Whatman filter paper grade 42, used explicitly for every cyanide solution sample, followed by solution filtering, recovery of fine AC particles, drying at 25 °C, counting of the particle mass, and fire assay.

### 2.6. Thermodynamic Analysis

The thermodynamic parameters of adsorption (Δ*G*, Δ*H*, and Δ*S*) indicate the spontaneity and feasibility of the adsorption process [26], as well as the influence of temperature on adsorption. Generally, thermodynamic parameters are determined using the equation of Van’t Hoff as follows:(4)$${\ln K}_{eq}=-\frac{{∆H}^{{}^{\circ}}}{RT}+\frac{{∆S}^{{}^{\circ}}}{R}$$
where *R* is the universal gas constant (1.987207 Cal/mol K), *T* is the temperature (K), and *K*_eq_ is the equilibrium constant. Δ*G* can be obtained from any of the following equations:Δ*G* = Δ*H* − *T*Δ*S*(5)
Δ*G* = −*RT* ln*K*_eq_(6)

Usually, if the change in Gibbs energy (Δ*G*) is negative (Δ*G* < 0) or positive (Δ*G* > 0), the reaction releases and absorbs energy, respectively, and both situations are possible for photocatalytic reactions [27].

The adsorption heat (Δ*H*) at a constant surface coverage was calculated using the following Clausius–Clapeyron equation [28]:(7)$$\frac{{\mathrm{d ln}(C}_{eq})}{dT}=-\frac{{∆H}^{{}^{\circ}}}{RT^{2}}$$

The integration of Equation (7) yields the following formula:(8)$${\ln(C}_{eq})=-\left[\frac{{∆H}^{{}^{\circ}}}{R}\right]\frac{1}{T}+K$$
where *K* is a constant.

From Equation (5), the value of Δ*S* can be obtained as follows:(9)$$\Delta S=\frac{{∆H}^{{}^{\circ}}}{T}-\frac{{∆G}^{{}^{\circ}}}{T}$$

Δ*H* and Δ*S* are independent of temperature. The positive or negative values of Δ*H* represent endothermic and exothermic reactions, respectively. A positive value of Δ*S* reflects the affinity of the adsorbent toward the sorbate, greater randomness at the solid–liquid interface, a greater degree of freedom of the sorbate, and more favorable conditions for the adsorption process to occur. However, a negative Δ*S* implies a less active interface of the solid–liquid system, causing a reduction in adsorption. Meanwhile, the negative and positive values of Δ*G* reflect spontaneous and nonspontaneous adsorption, respectively. Thus, the thermodynamic feasibility of a reaction depends on Δ*G*. The correct estimation of the thermodynamic parameter is essential for delineating the adsorption process [29,30].

### 2.7. Thermodynamic Models

#### 2.7.1. Langmuir Isotherm

The assumptions of the Langmuir isotherm model include a monolayer surface cover and identical and equivalent surface sites with the same sorption activation energy for each molecule, resulting in homogeneous adsorption and any interaction between adsorbed species in the surface plane [31,32,33].

The following mathematical expression represents the Langmuir isotherm:(10)$$q_{e}=\frac{q_{max}K_{L}C_{e}}{1+K_{L}C_{e}}$$
where $q_{e}$ represents the amount of absorbate adsorbed per unit weight of the absorbent in equilibrium (mol/g) and *C*_e_ is the solute concentration in equilibrium (mol/L). The parameters $q_{max}$ and *K*_L_ are Langmuir constants. $q_{max}$ is represented as the maximum adsorption capacity of the monolayer, and *K*_L_ represents the binding energy or affinity parameter of the adsorption system. *C*_e_ is usually represented in mg/L and *q*_e_ in mg/g for the solute adsorption of an aqueous solution in a solid medium. *q*_max_ and *K*_L_ correspond to the units mg/g and L/mg, respectively [34].

The Langmuir isotherm parameters can be determined by plotting 1/*C*_e_ vs. 1/*q*_e_, where *q*_e_ is determined according to the following equation:(11)$$q_{e}=\frac{C_{0}-C_{e}}{W}V$$
where *q*_e_ is the number of mols absorbed per gram of absorbent, *C*_0_ is the initial concentration, *C*_e_ is the equilibrium concentration or final concentration, *W* is the absorbent mass, and *V* is the volume of the aqueous medium.

#### 2.7.2. Temkin Isotherm

The Temkin isotherm model considers the indirect interaction effects between the absorbed molecules based on the Langmuir isotherm model. It is assumed that the heat of absorption (Δ*H*) of all molecules in the layer decreases linearly with increasing surface coverage. When the coverage is zero, the absorption energy is at its maximum. The extremely high and low adsorbate concentration conditions are ignored, and the equation of this model is as follows [29]:(12)$$q_{e}=\frac{RT}{B_{Te}}\ln\left(K_{T}C_{e}\right)=\frac{RT}{B_{Te}}\ln\left(K_{T}\right)+\frac{RT}{B_{Te}}\ln\left(C_{e}\right)$$

In this equation, *K*_T_ and $B_{Te}$ are the Temkin isotherm constants. *K*_T_ is the equilibrium constant corresponding to the maximum adsorption energy and has the same dimension as *K*_L_.

The adsorption reaction of the Temkin isotherm model is identical to that of the Langmuir isotherm model.

#### 2.7.3. Freundlich Isotherm

The Freundlich isotherm model is usually used as an adsorption isotherm model to represent nonlinear adsorption, expressed as follows [35,36]:(13)$$q_{e}=K_{F}\cdot C_{e}^{\frac{1}{n_{F}}}$$
where *K*_F_ is the Freundlich isotherm constant whose dimensions are determined by *C*_e_ and *q*_e_ and 1/*n*_F_ is the heterogeneous factor and is nondimensional. If *n*_F_ =1, the Freundlich isotherm reduces to a linear isotherm. The Freundlich isotherm is considered an empiric adsorption model without a clear physical meaning.

Equation (13) can be arranged in linear form as follows:(14)$$logq_{e}=log\;K_{f}+\frac{1}{n}\;log\;C_{e}$$

The Freundlich isotherm parameters were determined by plotting log *C*_e_ against log *q*_e_.

### 2.8. Kinetic Analysis

Adsorption technology is one of the most important and widely used technologies because of its relatively simple design and operation, cost-effectiveness, and energy efficiency. This technology is widely applied in water waste treatment, renewable energy, and chemical production. Adsorption kinetics can be described using classical models such as pseudo-first-order (PFO) and pseudo-second-order (PSO) [37,38].

#### 2.8.1. PFO Model

Lagergren presented the following expression of the reaction of the PFO model for *n* = 1 [39,40]:(15)$$\frac{dq_{t}}{dt}=k_{1}(q_{e}-q_{t})$$
where *q*_e_ and *q*_t_ are absorbate quantities obtained using the absorbate mass in equilibrium and at any time *t* (min), respectively, and *k*_1_ (min^−1^) is the velocity constant of the equation of PFO.

Integrating the last Equation (15) for limit conditions (*t* = 0, *q*_t_ = 0, and *t* = *t*, *q*_e_ = *q*_t_), the following equation is obtained:(16)$$ln\left(q_{e}-q_{t}\right)=\ln q_{e}-\;k_{1}t$$

The above equation can be arranged nonlinearly as follows:(17)$$q_{t}=q_{e}\;(1-e^{{-k}_{1}t})$$

The parameter of the pseudo-first-order model is determined by plotting *t* against ln(*q*_e_ − *q*_t_), where *q*_t_ is determined as follows:(18)$$q_{t}=\frac{C_{0}-C_{e}}{W}V$$
where $q_{t}$ is the number of adsorbed mols per gram of the adsorbent at time *t*, *C*_0_ is the initial concentration, *C*_e_ is the equilibrium concentration or final concentration, *W* is the absorbent mass, and *V* is the volume of the aqueous medium.

#### 2.8.2. PSO Model

The proposed expression for the reaction model of adsorption of PSO is shown below for *n* = 2:(19)$$\frac{dq_{t}}{dt}=k_{2}(q_{e}-q_{t}{)}^{2}\;$$

Integrating the above Equation (19) for limit conditions (*t* = 0, *q*_t_ = 0, and *t* = *t*, *q*_e_ = *q*_t_) yields the following linear equation:(20)$$q_{t}=\frac{q_{e}^{2}\;k_{2}t}{q_{e}k_{2}t+1}$$
where $q_{e}$ (mg/g) and *q*_t_ (mg/g) are adsorbent quantities obtained by the equilibrium adsorbent mass at any time *t* (min), respectively, and *k*_2_ (g/mg min) is the velocity constant of the PSO equation.

The above equation can be arranged linearly as follows [41]:(21)$$\frac{t}{q_{t}}=\frac{1}{k_{2}q_{e}^{2}}+\frac{1}{q_{e}}\;t$$

The parameters of the PSO model are determined by plotting *t* against *t*/*q*_t_, where *q*_t_ is determined using Equation (20).

## 3. Results

### 3.1. Gold Adsorption in AC in Fine Particles

The adsorption tests for the thermodynamic and kinetic analysis with variations in particle sizes began with the SEM characterization at 350×–2000×. Internal pores were observed in the development of spaces for gold adsorption. Figure 1 shows the different particle sizes used in these tests: (a) 106 μm, 140 mesh; (b) 75 μm, 200 mesh; (c) 53 μm, 270 mesh; (d) 45 μm, 325 mesh; (e) 38 μm, 400 mesh.

Although the AC particle size distribution exhibited a considerable effect on its external surface area, a considerably smaller effect was observed on the specific surface area because of the enormous internal development of internal porous structures [12].

In practice, many other factors affect the selection of AC particle size, such as the following:AC separation from rich (Pregnant Leach Solution-PLS) or barren (After AC Adsorption Process) solutions becomes complicated, particularly when dealing with the presence of carbon in extremely small particle sizes (generally, carbon screening can be conducted from 0.7 mm to 0.8 mm in barren solutions);The fine carbon generated by attrition is susceptible to being lost because of its mass loss and the high exposure of its surface area. Further, the size reduction usually causes the AC particles to leave the ADR column circuit rapidly;Small carbon has a shorter fluidization rate than regular-sized carbon, which affects the development of the process under the design characteristics of the equipment (e.g., supernatant carbon between circuit columns).

Systems that contain carbon with wide size distributions may experience less of a difference in gold loading because of an effect called contact ion exchange. In this readily measurable effect, gold is transferred from carbon with a high gold loading to carbon with a low gold loading. This is achieved through direct contact with thin fills surrounding the carbon particle, with negligible gold passing into the bulk solution [12].

### 3.2. Thermodynamic Analysis of Gold Adsorption in AC Langmuir, Temkin, and Freundlich Isotherms

The adsorption isotherm models of Langmuir, Temkin, and Freundlich are widely used for modeling adsorption process information. Table 2 shows the results obtained from the experimentation of gold adsorption at different AC charges in the tested particle sizes.

The results demonstrate that the process exhibits enhanced control based on the Freundlich isotherm model because of its *R*^2^ values. The thermodynamic behavior of the gold adsorption process in AC is described in this model.

This model explains the monolayer and multilayer adsorption processes. In addition, it explains that an adsorbent possesses surfaces of various affinities, or adsorption on heterogeneous surfaces. *K*_F_ and *n* are Freundlich constants denoting the approximate indicators of adsorption capacity and adsorption intensity in the adsorption process, respectively [29].

When plotting log *C*_e_ vs. log *q*_e_, we obtained the value of *K*_F_ from the antilogarithm of the value of the intercept with ordinates and the value of *n* when dividing with 1/*m*, where *m* is the value of the slope. The Freundlich plots for the five particle sizes are shown in Figure 2.

The smallest value of 1/*n* (i.e., 0.1993) and the highest value of *n* (i.e., 5.0176) indicate an active interaction between AC and gold. Meanwhile, the *K*_F_ value does not show a particular behavior with decreasing particle size. The highest value of *K*_F_ was demonstrated for the particle size of 75 μm, among the other particle sizes tested. This, in turn, showed the equilibrium concentration of gold with the lowest concentration at the end of the test in the presence of 0.250 g of AC, as observed in Table 3, which shows the capacity and intensity of the adsorption process for this particle size.

Table 3 shows the experimental data of gold adsorption in AC for the determination of the Freundlich isotherm. The experimental data for the Langmuir and Temkin isotherms are attached to this document as Appendix A.

Table 4 shows the thermodynamic parameters obtained from the adsorption of gold in AC from the Freundlich isotherm determination (log *C*_e_ vs. log $q_{e}$).

The negative value of Δ*G* confirmed the feasibility of the adsorption process and the spontaneous nature of the gold adsorption process in AC particles [27,28,29,30]. Meanwhile, the negative value of Δ*H* indicated the exothermic nature of the process [42]. The negative value of Δ*S* reflected a lower grade of disorder in the solid–liquid interphase during gold adsorption in AC particles [43].

### 3.3. Kinetic Analysis of Gold Adsorption in AC Using PFO and PSO

Most studies from the past two decades used the classical rate laws of PFO and PSO to model their kinetic datasets. These two models have been applied to a wide variety of adsorption systems, from biomass to nanomaterials (as adsorbents) and heavy metals to pharmaceuticals (as adsorbates or contaminants) [38].

Table 5 shows the parameters obtained for the gold adsorption in different AC sizes for the determination of the PFO model (time vs. ln(*q*_e_ − *q*_t_)).

Table 6 shows the parameters obtained for the gold adsorption in different AC sizes for the determination of the PSO model (time vs. *t*/*q*_t_).

The results demonstrate a process with greater control under the PSO model because of the *R*^2^ values. Moreover, a diffusion-controlled process is better described by *K*_2_ than by *K*_1_. In this study, *K*_1_ or *K*_2_ was declared the best velocity constant as long as it gave the highest coefficient of determination (*R*^2^), together with the condition that *R*^2^ > 0.80 [39].

The values obtained from the velocity constant *K*_2_ were taken and plotted against the particle size in the AC dosages (0.01, 0.05, 0.125, and 0.25 g), as shown in Figure 3.

Figure 3 generally shows an increase in adsorption speed with decreasing particle size, with some exceptions. The adsorption speed was found in the test with 0.25 g of AC at a particle size of 45 μm. Similarly, we observed that the highest adsorption in the test with 0.01 g of AC was obtained at a particle size of 45 μm. According to the PSO gold adsorption rate results, on 45 μm it found a breakpoint on the logical explanation where the smallest particle size should present greater rates of adsorption. These results amplify the vision on this scale of particle size; the hypothesis for the performance of AC on 45 μm at 0.25 and 0.01 g on the process of adsorption is focused on the porosity and activated sites to increase the rate of gold adsorption, as shown in Figure 1d.

The tests with 0.05 and 0.125 g of AC showed an increased adsorption speed with respect to the decrease in size.

### 3.4. Industrial Monitoring and Sampling of Escaping Fine AC Particles

To make an economic statement of the value of gold loss on AC fine particles in the industry, which would validate our scientific research as a worldwide and current problem, we requested authorization from the mining company Molimentales del Noroeste SA de CV to analyze their ADR plant for fine AC particle loss due to the dragging of the solution. Its plant C of ADR processes 960 m^3^/h of cyanide solution of sodium with an initial gold concentration of 0.080 mg/L through a train of six columns loaded with 2 tons of AC with a final gold concentration of 0.001 mg/L. Table 7 shows the results of the characterization of the cyanide solution analysis of the ADR Molimentales plant (density 1.004 g/mL).

The results obtained are presented for each of the solutions containing AC particles, indicating possible adsorbed gold content. A total of nine samples were randomly selected and evaluated using the fire assay technique to specifically determine the gold concentration, as shown in Table 8.

The results show gold concentration values of up to 74.532 g/T in the particles obtained during the 24-h sampling. Because the results also exhibit negative values, we averaged the gold content obtained from the nine samples tested. From the obtained values, we approximated the economic value of the lost gold due to the leakage of AC particles in the mining unit, and the analysis is presented in Table 9.

For the Molimentales del Noroeste mining company, for the sampling made in November 2022, an annual loss of 316,263.66 USD was determined from gold values lost in AC particles that escaped the process of the ADR plant.

However, the silver values and the secondary effects that the AC particles can have in the leaching grounds, owing to the preg-robbing effect, were not considered.

Technological developments have been patented and are being patented to recover AC particles that are lost worldwide owing to their considerable economic value and associated benefits. However, until now, the industry has yet to establish a definitive method.

## 4. Discussion

The characterization using SEM (Figure 1) shows a great adsorption capacity due to the increase in surface area. This was also because the 75 μm particles exhibited greater gold adsorption capacity than those of smaller fractions at 0.250 g of AC.

Bergna (2018) mentioned that different particle sizes in the range considered for his experiment did not affect the porosity characteristics of his samples, indicating that the activation process was a surface reaction. The results indicated that different particle sizes of starting biomass have a small but measurable influence on AC porosity [44]. This indicates the need to optimize the surface-volume relationship to improve the activated material.

Thus, size has a great effect on the average length of pores within carbon particles, and the adsorption ratio increases with decreasing particle size. This is an important factor at the industrial scale of adsorption systems because most operations work below the true load balance of gold in relation to the carrying capacity of AC [12].

SEM images (Figure 1a–d) indicate enormous internal development of porous structures. Figure 1e shows small diameters and a hard surface or less porosity on a plane surface of an AC particle at 38 μm, 400 mesh. Thus, the gold adsorption capacity is affected, as reflected in the AC characterization using fire assay and gravimetry in Table 3 under the equilibrium concentration parameter at 0.250 g of AC.

Müller and Gubbins (1998, p. 1435) [45] found that although the number of porosities increases by increasing the contact surface caused by the reduction of the particle size, it generates an adverse effect on the hydrophobicity of AC. Their results indicate that for a site density of approximately 0.5–2 nm, the selectivity of water increases relative to that of a carbon-free site by a factor of about 104. Even a small density of oxygenated sites on such carbons leads to substantial water adsorption, with the consequent loss of free surfaces for the adsorption of other components.

Marsden and House (2006, pp. 297–333) [12] mentioned that as the AC particle sizes increase, the adsorption capacity increases. Herein, our results showed that under the conditions described in this study and as shown in Figure 2, there are exceptions to what was mentioned in the aforementioned studies. That is, particle sizes other than the smaller dimension yield higher adsorption speeds and load capacity. We hypothesize that upon reaching a certain size, the particles exhibit hard and flat surfaces that hinder the adsorption process, as shown in Figure 1e.

The thermodynamic values of adsorption on AC, Δ*G* equal to −2.022 kcal/mol, Δ*H* equal to −16.710 kcal/mol, and Δ*S* equal to −0.049 kcal/molK were determined by the Freundlich model due the greatest affinity, according also to the study of Karnib et al. (2014) [23], since of the three models used Langmuir, Temkin, and the before mentioned for the equilibrium analysis, the Freudlich model reaches a coincidence value R^2^ > 0.85 in each of the tests carried out, the test with the particle size of 106 μm being the one with the lowest value (R^2^ = 0.8599) and the test with the size of 38 μm particle the one with the highest value (R^2^ = 0.9843), the latter being the test that reached the highest R^2^ of the three models, for which the values of slope, interception and constant (KF) were considered for the calculations of the thermodynamic parameters shown in this investigation.

## 5. Conclusions

This study demonstrated the magnitude of the impact of fine particles in ADR processes, both in their thermodynamic and kinetic aspects, as well as the economic aspect of their loss. Future research should be conducted to demonstrate the impact of AC fine particle placement at subsequent points of the metallurgical process, particularly in leaching pads (where they can cause the preg-robbing effect) and damage quantification.

The Freundlich thermodynamic model turned out to be the model where our gold adsorption experimental data for different particle sizes and weights were fitted, with R^2^ > 0.85. The maximum adsorption capacity of the adsorbent obtained was in the presence of 250 mg of AC particles at 75 μm after 120 min, due to the equilibrium concentration of 0.155 mg/L of gold.

The equilibrium data have been analyzed by Langmuir, Temkin, and Freundlich isotherm models. This Freundlich model shows that the adsorption process was physisorption. The charge in Gibbs free energy was negative, which confirmed the feasibility of the adsorption process and the spontaneous nature of the gold adsorption process in AC particles. The negative values of Δ*H* and Δ*S* indicated the exothermic nature of the process and reflected a lower grade of disorder in the solid–liquid interphase during gold adsorption in AC particles.

Kinetic studies showed that the adsorption adhered to the pseudo-second-order model, where the faster adsorption constant was in the presence of 250 mg of AC particles at 45 μm. Theoretical and experimental sorption capacities were in excellent agreement (R^2^ > 0.95).

The relationship between adsorption speed and particle size is subject to varying interpretations. In addition, future research is necessary to confirm the findings presented in this study, particularly due to the observed deviation from the proposed hypothesis in two out of the four experiments conducted.

The economic impact due to the loss of AC particles with gold content is directly related to the operating conditions of the metallurgical process. In this case study, USD 316,263.66 must be contextualized before any technological possibilities for its recovery in terms of the cost-benefit of the investment without neglecting that these particles impact other stages of the processes currently used worldwide.

## Figures and Tables

**Figure 1 materials-16-04961-f001:**
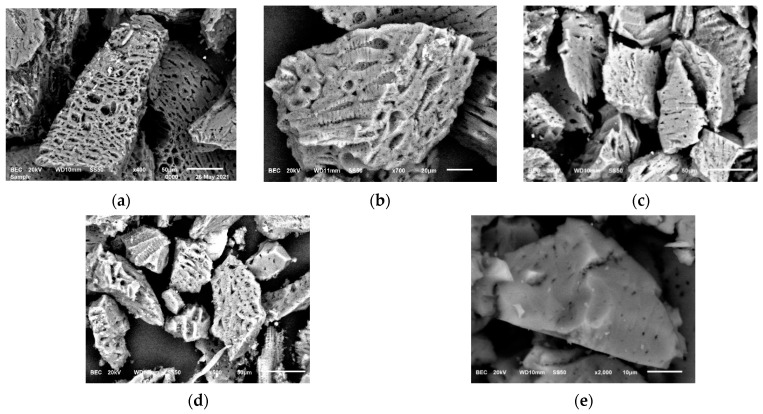
Characterization using a scanning electron microscope of AC used in gold adsorption tests: (**a**) 106 μm, 140 mesh; (**b**) 75 μm, 200 mesh; (**c**) 53 μm, 270 mesh; (**d**) 45 μm, 325 mesh; and (**e**) 38 μm, 400 mesh.

**Figure 2 materials-16-04961-f002:**
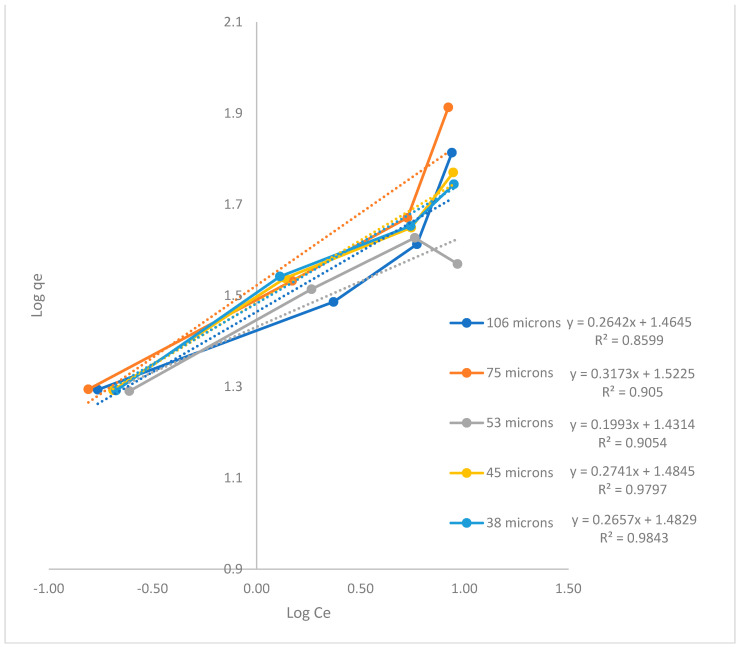
Gold adsorption graphs in the AC Freundlich isotherm model (log *C*_e_ vs. log *q*_e_).

**Figure 3 materials-16-04961-f003:**
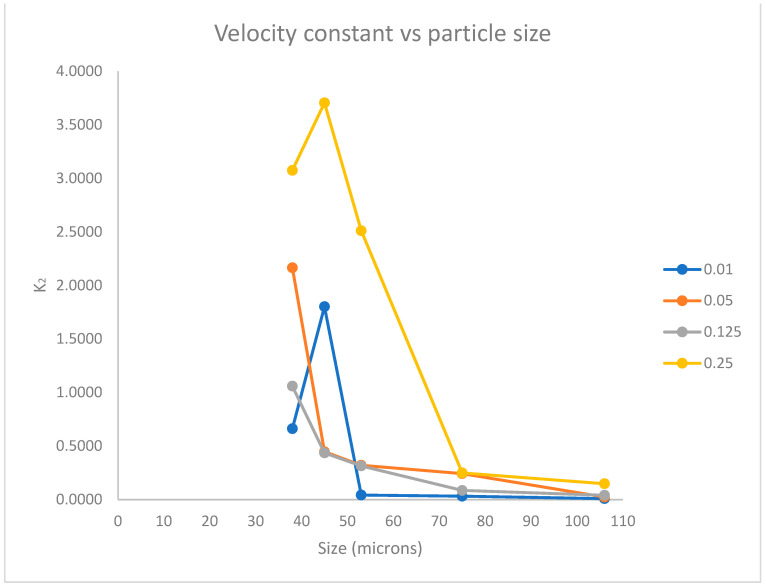
Graphic of different AC-sized particles vs. the gold adsorption rate of the PSO model with 0.01, 0.05, 0.125, and 0.25 g of AC.

**Table 1 materials-16-04961-t001:** Experimental conditions for the determination of the load capacity and adsorption rate of AC.

ID	Particle Size (μm)	Activated Carbon (mg)	Gold on Solution (mg/L)	Volume (L)
106-1	106	10	10	0.5
106-2		50		
106-3		125		
106-4		250		
75-1	75	10	10	0.5
75-2		50		
75-3		125		
75-4		250		
53-1	53	10	10	0.5
53-2		50		
53-3		125		
53-4		250		
45-1	45	10	10	0.5
45-2		50		
45-3		125		
45-4		250		
38-1	38	10	10	0.5
38-2		50		
38-3		125		
38-4		250		

**Table 2 materials-16-04961-t002:** Langmuir, Temkin, and Freundlich isotherm parameters.

**Particle Size** **μm**	**Langmuir Parameters**					
	**b**	**m**	**q_max_ (mg/g)**	**K_L_**	**R_L_**	**R^2^**
106	0.0228	0.0049	43.8596	4.6531	0.0210	0.8175
75	0.0191	0.0050	52.3560	3.8200	0.0255	0.8706
53	0.0253	0.0064	39.5257	3.9531	0.0247	0.9737
45	0.0205	0.0063	48.7805	3.2540	0.0298	0.9499
38	0.0205	0.0065	48.7805	3.1538	0.0307	0.9679
**Particle Size** **μm**	**Temkin Parameters**					
	**b**	**m**	**B_T_ (J/mol)**	**K_T_ (L/mg)**		**R^2^**
106	32.077	9.2522	9.2522	32.039		0.709
75	38.168	12.816	12.816	19.651		0.742
53	28.396	5.6752	5.6752	148.937		0.885
45	33.211	9.4598	9.4598	33.473		0.934
38	32.905	8.9616	8.9616	39.322		0.971
**Particle Size** **μm**	**Freundlich Parameters**					
	**b**	**m**	**1/n**	**K_F_**		**R^2^**
106	1.4645	0.2642	0.2642	29.1407		0.8599
75	1.5225	0.3173	0.3173	33.3043		0.9050
53	1.4314	0.1993	0.1993	27.0023		0.9054
45	1.4845	0.2741	0.2741	30.5141		0.9797
38	1.4829	0.2657	0.2657	30.4018		0.9843

**Table 3 materials-16-04961-t003:** Freundlich isotherm experimental data.

Particle Size	Activated Carbon Mass	*C_i_*	*C* _e_	1/*C*_e_	Log *C*_e_	Ln *C*_e_	*q* _e_	1/*q*_e_	Log *q*_e_
μm	g	mg/L	mg/L						
106	0.010	10	8.698	0.115	0.939	2.163	65.100	0.015	1.814
	0.050		5.903	0.169	0.771	1.775	40.970	0.024	1.612
	0.125		2.345	0.426	0.370	0.852	30.620	0.033	1.486
	0.250		0.172	5.814	−0.764	−1.760	19.656	0.051	1.293
75	0.010	10	8.363	0.120	0.922	2.124	81.850	0.012	1.913
	0.050		5.315	0.188	0.726	1.671	46.850	0.021	1.671
	0.125		1.483	0.674	0.171	0.394	34.068	0.029	1.532
	0.250		0.155	6.452	−0.810	−1.864	19.690	0.051	1.294
53	0.010	10	9.258	0.108	0.967	2.225	37.100	0.027	1.569
	0.050		5.766	0.173	0.761	1.752	42.340	0.024	1.627
	0.125		1.837	0.544	0.264	0.608	32.652	0.031	1.514
	0.250		0.244	4.098	−0.613	−1.411	19.512	0.051	1.290
45	0.010	10	8.822	0.113	0.946	2.177	58.900	0.017	1.770
	0.050		5.544	0.180	0.744	1.713	44.560	0.022	1.649
	0.125		1.369	0.730	0.136	0.314	34.524	0.029	1.538
	0.250		0.204	4.902	−0.690	−1.590	19.592	0.051	1.292
38	0.010	10	8.890	0.112	0.949	2.185	55.500	0.018	1.744
	0.050		5.505	0.182	0.741	1.706	44.950	0.022	1.653
	0.125		1.294	0.773	0.112	0.258	34.824	0.029	1.542
	0.250		0.211	4.739	−0.676	−1.556	19.578	0.051	1.292

**Table 4 materials-16-04961-t004:** Freundlich isotherm (log *C*_e_ vs. log *q*_e_) parameters.

	kcal/mol
Δ*G*	−2.022
Δ*H*	−16.710
	kcal/molK
Δ*S*	−0.049

**Table 5 materials-16-04961-t005:** PFO model results (time vs. ln(*q*_e_ − *q*_t_)).

Test ID	Parameters				
	b	m	*q*_e_ (mg/g)	K_1_	R^2^
106-1	1.7152	−0.0354	5.5578	−0.0003	0.9017
106-2	1.1155	−0.0433	3.0511	−0.0004	0.9846
106-3	0.7399	−0.0424	2.0957	−0.0004	0.9841
106-4	0.0136	−0.0617	1.0137	−0.0005	0.9849
75-1	0.5278	0.0300	1.6952	0.0003	0.9074
75-2	0.2819	−0.0687	1.3256	−0.0006	0.9547
75-3	0.3588	−0.0479	1.4316	−0.0004	0.9239
75-4	0.2776	−0.1226	1.3200	−0.0010	0.9967
53-1	0.6175	−0.0318	1.8543	−0.0003	0.8695
53-2	−0.8377	−0.0262	0.4327	−0.0002	0.7043
53-3	−0.4717	−0.0625	0.6239	−0.0005	0.9674
53-4	−2.6544	−0.0376	0.0703	−0.0003	0.7790
45-1	0.5532	−0.3230	1.7388	−0.0027	0.7484
45-2	−1.3866	−0.0212	0.2499	−0.0002	0.4183
45-3	−0.5542	−0.0719	0.5745	−0.0006	0.8980
45-4	−1.7486	−0.1471	0.1740	−0.0012	0.9478
38-1	−0.0658	−0.0201	0.9363	−0.0002	0.3851
38-2	−1.1021	−0.1505	0.3322	−0.0013	0.7090
38-3	−1.9300	−0.0159	0.1451	−0.0001	0.3002
38-4	−2.7675	−0.0441	0.0628	−0.0004	0.7318

**Table 6 materials-16-04961-t006:** PSO model (time vs. *t*/*q*_t_).

Test ID	Parameters					
	b	m	*q*_e_ (mg/g)	*q*_e_^2^ (mg/g)	K_2_	R^2^
106-1	2.0474	0.1402	7.1327	50.8749	0.0096	0.9760
106-2	2.1641	0.2255	4.4346	19.6656	0.0235	0.9904
106-3	2.3488	0.3078	3.2489	10.5551	0.0403	0.9978
106-4	1.6362	0.4926	2.0300	4.1211	0.1483	0.9998
75-1	0.1928	0.1217	8.2169	67.5179	0.0768	0.9997
75-2	0.4896	0.2108	4.7438	22.5040	0.0908	0.9996
75-3	0.8691	0.2863	3.4928	12.1999	0.0943	0.9991
75-4	0.5303	0.5025	1.9900	3.9603	0.4762	0.9999
53-1	1.5994	0.2608	3.8344	14.7023	0.0425	0.9956
53-2	0.1746	0.2366	4.2265	17.8637	0.3206	0.9999
53-3	0.2939	0.3038	3.2916	10.8349	0.3140	0.9999
53-4	0.1045	0.5122	1.9524	3.8117	2.5105	1.0000
45-1	0.0164	0.1719	5.8173	33.8414	1.8018	0.9984
45-2	0.1135	0.2251	4.4425	19.7355	0.4464	0.9997
45-3	0.1906	0.2881	3.4710	12.0480	0.4355	1.0000
45-4	0.0701	0.5096	1.9623	3.8507	3.7046	1.0000
38-1	0.0491	0.1803	5.5463	30.7616	0.6621	0.9990
38-2	0.0226	0.2212	4.5208	20.4376	2.1650	0.9998
38-3	0.0792	0.2897	3.4518	11.9152	1.0597	0.9996
38-4	0.0847	0.5103	1.9596	3.8402	3.0745	1.0000

**Table 7 materials-16-04961-t007:** Solution analysis of the ADR Molimentales plant (density 1.004 g/mL).

ID	Solution	AC Mass	Solution	AC on Solution
	(g)	(g)	(mL)	(g/L)
1	887.25	0.0077	883.41	0.008716
2	879.80	0.0012	876.00	0.001370
2 RPT	891.36	0.0024	887.51	0.002704
3	894.09	0.0051	890.23	0.005729
3 RPT	887.66	0.0154	883.82	0.017424
4	901.58	0.0208	897.69	0.023171
4 RPT	869.93	0.0173	866.17	0.019973
5	873.98	0.0047	870.20	0.005401
6	876.40	0.0147	872.61	0.016846
6 RPT	844.19	0.0267	840.54	0.031765
7	847.53	0.0338	843.87	0.040054
8	884.74	0.0250	880.92	0.028380
8 RPT	854.34	0.0616	850.65	0.072415
9	880.69	0.0226	876.89	0.025773
9 RPT	853.78	0.0260	850.09	0.030585
10	869.72	0.0233	865.96	0.026906
11	852.50	0.0243	848.82	0.028628
12	886.68	0.0225	882.85	0.025486
12 RPT	864.64	0.0273	860.90	0.031711
13	890.70	0.0282	886.85	0.031798
14	877.61	0.0257	873.82	0.029411
15	865.24	0.1064	861.50	0.123506
15 RPT	884.22	0.1054	880.40	0.119719
16	889.00	0.0102	885.16	0.011523
17	883.62	0.0024	879.80	0.002728
18	919.10	0.0052	915.13	0.005682
18 RPT	902.46	0.0001	898.56	0.000111
19	926.68	0.0035	922.68	0.003793
20	871.09	0.0048	867.33	0.005534
21	910.70	0.0013	906.77	0.001434
21 RPT	879.24	0.0026	875.44	0.002970
22	856.50	0.0021	852.80	0.002462
23	859.84	0.0022	856.13	0.002570
24	882.99	0.0030	879.18	0.003412
24 RPT	826.65	0.0017	823.08	0.002065
24 RPT	862.25	0.0029	858.53	0.003378

**Table 8 materials-16-04961-t008:** Gold analysis in AC particle recovery from the ADR Molimental plant.

Test Number	ID	Mass (g)	Gold (g/T)
1	4	0.021	0.000
2	6-2	0.027	74.532
3	8-2	0.062	25.325
4	9-2	0.026	0.000
5	12-2	0.027	31.136
6	15-2	0.105	56.926
7	18	0.005	23.077
8	21-2	0.003	0.000
9	24-1	0.003	0.000
MEAN			26.374

**Table 9 materials-16-04961-t009:** Economic analysis of AC particles recovered from the ADR Molimentales plant.

Molimentales del Noroeste SA de CV (2022)					
Avarege Flow on ADR Plant	AC Fine Particle Loss	Gold on AC Fine Particle	Gold Loss	Gold Loss	USD (Oz of Au $2008.02)
m^3^/h	T/Year	g/T	g/Year	oz/Year	Dollars
960	185.74	26.37	4898.79	157.50	$316,263.66

## Data Availability

Not applicable.

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
