# Peer review of "Thermodynamic and Kinetic Aspects of Gold Adsorption in Micrometric Activated Carbon and the Impact of Their Loss in Adsorption, Desorption, and Reactivation Plants"

_materials, 2023, doi:10.3390/ma16144961_

Round 1

Reviewer 1 Report

1- The manuscript needs to be thoroughly revised.

2- Abstract needs to be thoroughly amended. Half of the abstract is an introduction; the research problem is not well defined; Need to describe what you have done in the manuscript; The results statement is too long and there are no specific conclusions. Change “obtains” in Abstract, line 19 to “gains”

3- In the introduction, “ NaCN is one the main drivers of gold 34 dissolution, but it…… Elsner equation [5,6], as shown in the following:” please omit “one”, replace “as shown in the following:” by “as follows.”

4- The research gap/problem and the contribution of the paper are not clear.

5- The conclusions of the study are not clear.

 6- English language needs significant improvements. Despite there are no grammatical or spelling mistakes, sentences structures are very hard to follow and understand. Sentences are too long; some of them extends to 6 lines.

Some sentences are not scientific. 

Some sentences structure are hard to follow and understand.

Author Response

We appreciate all your comments. We want to let you know that our research was split into two articles due to the extension of the goal and the results achieved, the article that we submitted to this Journal covers the importance of the losses of fine particles and the economic impact on ADR plants. The second one was submitted to Hydrometallurgy Journal from Elsevier (*see citation), it is under review and explains our results about recovering AC fine particles using an electrocoagulation process, with positive results.

Citation of Second Article:

Martinez Peñuñuri, Rodrigo and Parga-Torres, J. R. and Valenzuela-García, Jesús Leobardo and García-Alegría, A. M. and González-Zamarripa, G., Recovery of Fine Particles of Activated Carbon with Gold by the Electrocoagulation Process Using a Taguchi Experimental Design. Available at SSRN: https://ssrn.com/abstract=4213228 or http://dx.doi.org/10.2139/ssrn.4213228

1- The manuscript needs to be thoroughly revised.

The manuscript was modified following the suggestions of the three reviewers, and also, was sent to our editorial partner Eango, in order to do an extended revision.

2- Abstract needs to be thoroughly amended. Half of the abstract is an introduction; the research problem is not well defined; Need to describe what you have done in the manuscript; The results statement is too long and there are no specific conclusions. Change “obtains” in Abstract, line 19 to “gains”

R.- The abstract was changed in order to define better the problem, methodology, and results. Lines 15-30

3- In the introduction, “NaCN is one the main drivers of gold 34 dissolution, but it…… Elsner equation [5,6], as shown in the following:” please omit “one”, replace “as shown in the following:” by “as follows.” 

R.- It was erased “one”, and changed as shown in the following by “as follows”. Line 41

4- The research gap/problem and the contribution of the paper are not clear.

R.- The abstract, some specific comments, and conclusions were changed in order to define better the problem, methodology, and results.

5- The conclusions of the study are not clear.

R.- The conclusions were modified in order to make them clear. Lines 531-560

 6- English language needs significant improvements. Despite there are no grammatical or spelling mistakes, sentences structures are very hard to follow and understand. Sentences are too long; some of them extends to 6 lines.

Comments on the Quality of English Language

Some sentences are not scientific. 

Some sentences structure are hard to follow and understand.

R.- Our Editorial partner Enago improves the use of English

Reviewer 2 Report

This work studied the impact of fine AC particles in ADR processes through thermodynamic and kinetic aspects, as well as the economic aspect. The authors collected a lot of data to conduct this study. In my opinion, this work can be published after revision:

1. It would be better if the authors discuss how the data are used to calculate the related thermodynamic functions (ΔH, ΔG, ΔS) in the results section.

2. The grid lines in Figures can be deleted. 

The quality of english can be improved.

Author Response

We appreciate all your comments. We want to let you know that our research was split into two articles due to the extension of the goal and the results achieved, the article that we submitted to this Journal covers the importance of the losses of fine particles and the economic impact on ADR plants. The second one was submitted to Hydrometallurgy Journal from Elsevier (*see citation), it is under review and explains our results about recovering AC fine particles using an electrocoagulation process, with positive results.

Citation of Second Article:

Martinez Peñuñuri, Rodrigo and Parga-Torres, J. R. and Valenzuela-García, Jesús Leobardo and García-Alegría, A. M. and González-Zamarripa, G., Recovery of Fine Particles of Activated Carbon with Gold by the Electrocoagulation Process Using a Taguchi Experimental Design. Available at SSRN: https://ssrn.com/abstract=4213228 or http://dx.doi.org/10.2139/ssrn.4213228

  1. It would be better if the authors discuss how the data are used to calculate the related thermodynamic functions (ΔH, ΔG, ΔS) in the results section.

We add on discussion section the next comment:

The thermodynamic values of adsorption on AC, ΔG equal to -2.022 kcal/mol, ΔH equal to -16.710 kcal/mol, and ΔS equal to -0.049 kcal/molK were determined by the Freundlich model due the greatest affinity, according also to the study of Karnib et al (2014) [23], since of the three models used Langmuir, Temkin, and the before mentioned for the equilibrium analysis, the Freudlich model reaches a coincidence value R2 > 0.85 in each of the tests carried out, the test with the particle size of 106 μm being the one with the lowest value (R2 = 0.8599) and the test with the size of 38 μm particle the one with the highest value (R2 = 0.9843), the latter being the test that reached the highest R2 of the three models, for which the values of slope, interception and constant (KF) were considered for the calculations of the thermodynamic parameters shown in this investigation. Lines 519-529

  1. The grid lines in Figures can be deleted. 

R.- The grid lines in Figures were deleted

Comments on the Quality of English Language

The quality of english can be improved.

R.- Our Editorial partner improves the use of English

Reviewer 3 Report

Represented paper is dedicated to the problem of the increasing of gold extraction efficiency from cyanide solutions by adsorption on active carbons by avoiding the loss of AC particles smaller than 0.4 mm.  This problem is undoubtedly of interest to industry, but in general it looks like a purely applied and doesn’t provide new results with scientific importance. From a scientific point of view, it would be wonderful, for example, if the authors proposed some ideas how to improve the rigidity and strength of the adsorbent.

Several comments:

1.       Line 66. Please give a more detailed explanation for the value of AC consumption in ADR processes: 40 g (of what?)/ton (of what?).

2.       Section 2.1. Please provide more information about the active carbon: origin, activation type, etc. It is also recommended to give some information about its porous structure and the reasons of its selection.

3.       Sections 2.4, 3.2. An additional explanation for adsorption measurement procedure should be provided (e.g. temperature) as well as the reason for the study of adsorption on the particles of all sizes. Obviously, a small difference in the sizes of the particles (38-106 microns) cannot significantly affect the equilibrium adsorption. This is evidenced by obtained results shown in the Fig.2 (it is recommended to plot all isotherms on one graph). At the same time the difference can occur in comparison with the particles with higher dimensions – above 1 mm.

4.       It is recommended to reduce the sections 3.2-3.3 by deleting useless information in the tables with the data on different approximation methods…

5.       Lines 87-89. Please provide a reference to support the statement that the particles smaller than 0.4 mm usually escape.

6.       Lines 158-159. Elemental EDX analysis is also claimed to have been performed, but the results are not reported in the text.

7.       Table 1 can be optimized and reduced, because amounts of active carbon, gold in solution and volume are the same for all the samples.

8.       Section 2.5. Does this industrial process use the same active carbon, solutions etc.?

9.       Figure 1. If authors wished to highlight the difference between the sizes of the particles, the scale should be the same for all the figures, however the difference between will be small. If it is aimed at demonstrating macro-porosity, it would be enough to leave three pictures e.g. a, b, e.

10.   Table 4. Cal and Kcal – is it really necessary to give two values? Entropy unit is invalid.

11.   Figure 3 needs more explanations on occurring maximal points for the “45 microns” AC. Curve “0.01” – please give an explanation for negative value of K2 for the particles ranging in size from 55 to 70 microns.

12.   Section 3.4. This section looks like a separate part of the article, not connected in any way with the work described above.

13.   Section 4 is very short, taking in account huge volume of the paper. It is recommended to combine it with the conclusions of section 5, since the latter mainly is related to section 3.4...

Extensive editing of English language required

Author Response

We appreciate all your comments. We want to let you know that our research was split into two articles due to the extension of the goal and the results achieved, the article that we submitted to this Journal covers the importance of the losses of fine particles and the economic impact on ADR plants. The second one was submitted to Hydrometallurgy Journal from Elsevier (*see citation), it is under review and explains our results about recovering AC fine particles using an electrocoagulation process, with positive results.

Citation of Second Article:

Martinez Peñuñuri, Rodrigo and Parga-Torres, J. R. and Valenzuela-García, Jesús Leobardo and García-Alegría, A. M. and González-Zamarripa, G., Recovery of Fine Particles of Activated Carbon with Gold by the Electrocoagulation Process Using a Taguchi Experimental Design. Available at SSRN: https://ssrn.com/abstract=4213228 or http://dx.doi.org/10.2139/ssrn.4213228

Several comments:

  1.      Line 66. Please give a more detailed explanation for the value of AC consumption in ADR processes: 40 g (of what?)/ton (of what?).

R.- It was clarified what the concentration of 40 g/ton corresponds to (40 grams of AC per ton of ore processed) Line 72

  1.      Section 2.1. Please provide more information about the active carbon: origin, activation type, etc. It is also recommended to give some information about its porous structure and the reasons of its selection.

R.- In section 2.1 more information was provided about activated carbon: origin, type, pore structure, the reason for its selection, etc. Lines 109-119

  1.      Sections 2.4, 3.2. An additional explanation for adsorption measurement procedure should be provided (e.g. temperature) as well as the reason for the study of adsorption on the particles of all sizes. Obviously, a small difference in the sizes of the particles (38-106 microns) cannot significantly affect the equilibrium adsorption. This is evidenced by obtained results shown in the Fig.2 (it is recommended to plot all isotherms on one graph). At the same time the difference can occur in comparison with the particles with higher dimensions – above 1 mm.

R.- We add temperature, pressure, and conductivity parameters. We mention the goal of looking at the adsorption of the sizes selected to study. We change the graph as suggested. 177-186

  1.      It is recommended to reduce the sections 3.2-3.3 by deleting useless information in the tables with the data on different approximation methods…

R.- The information is considered relevant to the study, due to state the comparison between PFO and PSO tendency (R2), also the tables are a resume from all the information attached to the Appendix A

  1.      Lines 87-89. Please provide a reference to support the statement that the particles smaller than 0.4 mm usually escape.

R.- A bibliographical reference was included in support of these data, also 0.4 mm corresponds to the mesh that was used on the industrial operation that is mentioned in the article. Lines 90-93

  1.      Lines 158-159. Elemental EDX analysis is also claimed to have been performed, but the results are not reported in the text.

R.- The information provided by EDX in the results section was discarded, and the mention on the methodology section was erased.

  1.      Table 1 can be optimized and reduced, because amounts of active carbon, gold in solution and volume are the same for all the samples.

R.- Table 1 was modified according to the reviewer's suggestion.

  1.      Section 2.5. Does this industrial process use the same active carbon, solutions etc.?

R.- There was included a commentary about the same AC from Molimentales del Noroeste SA de CV according to the AC used in our research. Line 205

  1.      Figure 1. If authors wished to highlight the difference between the sizes of the particles, the scale should be the same for all the figures, however the difference between will be small. If it is aimed at demonstrating macro-porosity, it would be enough to leave three pictures e.g. a, b, e.

R.- We considered it important to maintain all the figures in Figure 1. Figure 1 (d) was mentioned in the article, also, our goal is to demonstrate the five different particles used in our research.

  1.  Table 4. Cal and Kcal – is it really necessary to give two values? Entropy unit is invalid.

R.- Table 4 was modified according to the reviewer's suggestion.

  1.  Figure 3 needs more explanations on occurring maximal points for the “45 microns” AC. Curve “0.01” – please give an explanation for negative value of K2 for the particles ranging in size from 55 to 70 microns.

R.- The previous graphic type was doing a tendency on Figure 3; it was changed due to negative values are not correct for the purpose of the study

  1.  Section 3.4. This section looks like a separate part of the article, not connected in any way with the work described above.

R.- Section 3.4 is introduced methodology in section 2.5, and shows an important part of the study, the relation between the scientific purpose of our research bond with the worldwide problem, and the validation of economic value.

  1.  Section 4 is very short, taking in account huge volume of the paper. It is recommended to combine it with the conclusions of section 5, since the latter mainly is related to section 3.4...

R.- Section 4 is a brief discussion of the main points of our research.

Comments on the Quality of English Language

Extensive editing of English language required

R.- Our Editorial partner Enago improves the use of English

Round 2

Reviewer 1 Report

The manuscript can be accepted.

Minor English editing is required.

Reviewer 3 Report

accept

Moderate editing of English language required